# Scaffold Guided Bone Regeneration for the Treatment of Large Segmental Defects in Long Bones

**DOI:** 10.3390/biomedicines11020325

**Published:** 2023-01-24

**Authors:** Frank Schulze, Annemarie Lang, Janosch Schoon, Georgi I. Wassilew, Johannes Reichert

**Affiliations:** 1Center for Orthopaedics, Trauma Surgery and Rehabilitation Medicine, University Medicine Greifswald, 17475 Greifswald, Germany; 2Departments of Orthopaedic Surgery & Bioengineering, University of Pennsylvania, Philadelphia, PA 19104, USA

**Keywords:** bone regeneration, growth factors, scaffolds, vascularization

## Abstract

Bone generally displays a high intrinsic capacity to regenerate. Nonetheless, large osseous defects sometimes fail to heal. The treatment of such large segmental defects still represents a considerable clinical challenge. The regeneration of large bone defects often proves difficult, since it relies on the formation of large amounts of bone within an environment impedimental to osteogenesis, characterized by soft tissue damage and hampered vascularization. Consequently, research efforts have concentrated on tissue engineering and regenerative medical strategies to resolve this multifaceted challenge. In this review, we summarize, critically evaluate, and discuss present approaches in light of their clinical relevance; we also present future advanced techniques for bone tissue engineering, outlining the steps to realize for their translation from bench to bedside. The discussion includes the physiology of bone healing, requirements and properties of natural and synthetic biomaterials for bone reconstruction, their use in conjunction with cellular components and suitable growth factors, and strategies to improve vascularization and the translation of these regenerative concepts to in vivo applications. We conclude that the ideal all-purpose material for scaffold-guided bone regeneration is currently not available. It seems that a variety of different solutions will be employed, according to the clinical treatment necessary.

## 1. Introduction

Bone possesses a considerable inherent regenerative potential. The treatment of large segmental bone defects, however, remains a clinical challenge. A variety of conditions related to segmental bone defects, defect size, localization, and microenvironment lead to an impaired healing. Such conditions include infections, defects after tumor resection, defects after revision arthroplasty, fractures associated with extensive soft tissue damage, and congenital deformities [1]. Usually, such defects require the restoration of a large amount of high-quality bone to achieve sufficient healing, which greatly depends on the presence of an adequate vascular supply to aid bone regeneration and remodeling.

The spectrum of the present surgical management of such defects is broad and includes amputation, limb shortening, non-vascularized autograft transplantation, staged bone grafting (Masquelet technique), distraction osteogenesis (Ilizarov technique) and vascularized grafts [2,3]. These strategies aim to provide an osteogenic or conductive stimulus to facilitate bone regeneration and union. However, they face certain limitations, such as tissue availability, donor site morbidity, varying levels of osteogenicity, late biomechanical failure and possible disease transmission [1]. Despite advances and growing experience with the application of bone graft substitutes and the Ilizarov technique, complex cases still pose a severe challenge and thus demand novel treatment strategies.

Large bone defect regeneration is often additionally impeded by a compromised wound healing environment, and necessitates concepts to both replenish the bone gap and to support vascularization and repair. Consequently, soft tissue reconstruction, utilizing free vascularized flaps, is considered an important approach that is successfully applied in clinical settings. These procedures are complex, time-consuming, associated with high costs, extended periods of hospitalization and often uncertain outcomes [4].

Clinically driven research for alternative therapeutic strategies has motivated multidisciplinary teams in the field of musculoskeletal tissue engineering to address these reconstructive challenges. Here, the underlying general strategy relies on the combination of osteoconductive scaffolds with osteogenic cells and growth factors, and suitable mechanical stimuli to create biomimetic bone substitutes [5]. Despite numerous advances over the last decade, a strategy for the successful, scaffold-based, routine clinical treatment of large segmental bone defects has yet to be found.

The present work reviews the strategies and technical approaches used to overcome the multilayered problems associated with large bone defect healing in long bones, with emphasis on research rooted in scaffold-guided tissue regeneration. In this context, large bone defects are referred to as defects of critical size that preclude spontaneous healing without intervention [6]. In case of intervention, the lost bone volume must specifically be addressed and a sufficient blood supply be provided, considering the absence of local endogenous osteogenic signals. With our review, we aim to provide an overview of the different components that form part of the proposed tissue engineering solutions. In addition, we want to provide insights into future approaches that aim to team the different elements for the successful translation of advanced therapies to routine clinical practice.

## 2. Physiology of Bone Healing

Bone is a biologically active tissue, characterized by a well-orchestrated balance of the osteoblast-mediated formation and osteoclast-driven resorption of extracellular matrix, which facilitates constant remodeling. Bone is known to be one of the few tissues that can heal without the formation of a scar and the processes involved in bone regeneration are considered to closely resemble well-conserved mechanisms of embryonic development. Bone healing can generally be subdivided into a defined chronologic order of events, which include a phase of inflammation, proliferation and differentiation, followed by tissue mineralization and remodeling. The single events taking place may, however, temporally overlap or vary locally within a site of regeneration [7].

After bony fracture and attendant vascular disruption, a local hematoma occurs; this is rich in bone- and blood-derived immunocompetent and osteogenic cells, platelets and macrophages [8]. The local release of proinflammatory cytokines, such as interleukin (IL)-1β, IL-6 and tumor necrosis factor-α (TNF-α), increase vascular permeability, promote cell migration and activate thrombocytes, as well as phagocytic effector cells [9,10]. This is amplified by local hypoxia that leads to the further disposal of cytokines from necrotizing tissue at the site of injury.

With subsiding inflammation, the reparative processes also involve new tissue formation and commencing granulation. Mediated by growth factors, such as transforming growth factor-β (TGF-β), bone morphogenic proteins (BMP), fibroblast growth factor (FGF), insulin like growth factor (IGF) and platelet derived growth factor (PDGF) [7], resident cells, such as mesenchymal stem cells (MSC), proliferate and differentiate into osteoblasts and chondrocytes. MSCs and their progeny produce and deposit extracellular matrix (ECM) proteins, such as collagen type I (Col I) and II (Col II), to form the soft tissue callus; this is gradually invaded by newly formed blood vessels [11]. This callus cannot provide sufficient mechanical stability until the gradual mineralization of the calcium-phosphate hydroxyapatite (HA) takes place, by incorporation into the ECM.

The development and formation of the skeleton (ossification) occurs by two distinct processes—intramembranous and endochondral ossification [7,12]. Both intramembranous and endochondral ossification occur in close proximity to vascular in-growth. Intramembranous ossification is characterized by the invasion of capillaries into the mesenchymal zone and the emergence and differentiation of mesenchymal cells into mature osteoblasts [13]. These osteoblasts initially proliferate and constitutively deposit bone matrix, leading to the formation of bone spicules. These spicules grow and develop, eventually fusing with other spicules to form trabeculae. As the trabeculae increase in size and number, they become interconnected and form woven bone, a disorganized weak structure with a high proportion of mechanosensitive osteocytes [14]. Intramembranous ossification occurs during embryonic development and is involved in the development of flat bones in the cranium, various facial bones, parts of the mandible and clavicle, and the addition of new bone to the shafts of most other bones [13].

In contrast, bones of load-bearing joints form by endochondral formation, which requires preformed cartilaginous tissue [8]. These cartilaginous intermediates are produced by chondrocytes, originating from differentiating MSCs that secrete a glycosaminoglycan and Col II-rich matrix. This template is subsequently degraded enzymatically and replaced by osseous tissue containing predominantly Col I [15].

Independent of its mode, ossification initially results in the formation of woven bone. This immature bone tissue is subsequently remodeled to form biomechanically competent, organized lamellar bone that is adapting its architecture to local mechanical needs. The process of remodeling in response to mechanical stimuli is mediated by the well-orchestrated activity of osteoblasts, osteocytes and osteoclasts [16].

Bone remodeling is characterized by a dynamic interplay of sequential mechanisms that lead to changes in EMC composition mediated by cell adhesion, proliferation and differentiation, and the release of active signaling molecules. In this context, matrix metalloproteinases (MMPs) play an important regulatory role, as they participate in various phases of bone regeneration [17].

The exact role of the different MMPs involved in bone remodeling is yet to be determined. MMP-2, however, appears to play a role in embryonic bone development, while MMP-7 stimulates bone maturation and ECM degradation, with MMP-9 driving osteoclast-mediated bone remodeling [18]. Generally, MMPs are secreted as inactive pro-enzymes or zymogens and require activation through proteolytic cleavage. MMP performance is modulated by their tissue inhibitors (TIMP 1-4) [18] and the post-transcriptional influence of specific miRNAs [19]. MMPs are secreted by mature osteocytes and locally degrade bone ECM to form the interconnected lacuna-canalicular system [20]. This network enables osteocytes to keep physical contact with the ECM. As a result, actin filaments and microtubules link together with crosslinking proteins, such as vinculin and fibrin; all of these form part of the cytoskeleton and receive mechanical and chemical stimuli that ultimately lead to a specific response [21]. Stimuli remitted to the cytoplasm lead to the activation of signaling pathways that are, for example, dependent on Cx43, MAPK/ERK, Wnt, YAP/TAZ, and Rho-ROCK-mediating cytoskeletal changes in osteocytes and ECM remodeling. A detailed description of the underlying mechanisms is beyond the scope of this article and is reviewed in detail elsewhere [17].

In addition to MMPs, osteocytes synthesize another family of ECM proteins, referred to as the small integrin-binding ligand, N-linked glycoprotein (SIBLING). SIBLING members, such as DMP-1, MEPE, FGF-23, PHEX and Fetuin-A, contribute to the regulation of the phosphocalcic metabolism and mineralization [22].

Bone collagen fibers are the major ECM component that confer resilience and tensile strength. Collagen fibers consist of fibril bundles formed by self-assembly. Extracellularly, collagen fibers are subject to extensive lysine-derived cross-linking to form stable polymers. The cross-linking is initiated by the enzyme, lysyl oxidase (LOX) [23]. LOX is a member of a copper amine oxidase family that includes LOX and LOX-like (LOXL) 1 to 4. The LOX gene is firmly regulated during development. LOX protein activation critically depends on Cu concentrations and redistribution. In the ECM, the pro-LOX is proteolytically activated by procollagen C-proteinases into an enzyme and a pro-peptide fragment (LOX-PP). The LOX-PP has a range of well-characterized functions, which are independent of LOX and are primarily associated with carcinogenesis [24].

Aberrant LOX expression patterns and functions have been associated with a multitude of diseases, such as vascular, cardiac, pulmonary, dermal, placenta, diaphragm, kidney and pelvic floor disorders, glioblastoma, diabetic neovascularization, osteogenic differentiation and bone matrix formation, ligament remodeling, polycystic ovary syndrome, fetal membrane rupture, and stages of tumor progression and metastasis in different entities of cancer. These processes usually involve LOX interaction with the signal transduction of multiple regulatory pathways that are, for example, associated with the epidermal growth factor receptor (EGFR), platelet derived growth factor (PDGF), vascular endothelial growth factor (VEGF), transforming growth factor β (TGF-β), ERK, NF-κB, PI3K/AKT, SMAD, MAPK, FAK/AKT, inflammation and steroid regulation [25].

The fibulins are yet another family of matricellular proteins without a structural role. Fibulin-4, and its paralog fibulin-5 protein, has been shown to be indispensable for elastic fiber assembly. There is also evidence for its regulatory involvement in skeletal development and bone stability due to its abnormal collagen fibers, suggesting functional links between fibulin-4 and LOX [23,26].

## 3. Tissue Engineering and Regenerative Medicine

### 3.1. Scaffolds for Bone Tissue Engineering

#### 3.1.1. Requirements

Scaffold-based biomimetic bone substitutes are meant to replace missing tissue by mimicking the structural, mechanical and biological properties of bone. In general, bone substitutes, applied for the treatment of large segmental defects, need to promote osteoinduction, osteoconduction and osseointegration. Osteoinduction describes the ability to induce osteogenesis by stimulating the differentiation of pluripotent precursor cells into bone-forming osteoblasts [27,28]. The ability to facilitate growth on a scaffold surface, and facilitate in-growth into pores or channels by cell adherence, proliferation and formation of a new ECM, is referred to as osteoconductivity [29]. The proper formation of a mechanically stable direct contact between bone tissue and implanted material, without the growth of fibrous tissue, is defined as osseointegration [30]. To meet these requirements, a number of scaffold properties, such as material composition and spatial organization, need to be considered and carefully balanced. Materials that are used as scaffolds in regenerative approaches need to exhibit sufficient biocompatibility. Biocompatibility can be defined as ‘the ability of a biomaterial to perform its desired function with respect to a medical therapy, without eliciting any undesirable local or systemic effects on the recipient or beneficiary of the therapy, but meanwhile generating the most optimized clinically relevant performance’ [31]. This relates to both the bulk form of a given material and the possible degradation products. In detail, the biocompatibility of a scaffold material involves the promotion of cell survival and the preservation of cellular functions specific for a given phenotype, while avoiding the induction of cell apoptosis or immunogenic responses [32,33]. In contrast to restorative methods, such as joint replacement, regenerative approaches require scaffold biodegradability, since the ultimate goal here is to stimulate and promote endogenous tissue healing [34]. Consequently, an ideal scaffold material for tissue regeneration is completely degradable over time and gradually replaced by endogenously formed bone matrix. At the same time, the scaffold needs to provide adequate mechanical support. Hence, material properties, such as compressive strength, stiffness and elasticity, should mimic those reported as characteristic for bone at the respective stage of regeneration [35,36,37]. For example, the biomechanical environment during endochondral ossification is characterized by a rather low Young’s modulus of approximately 8 kPa, while that value is situated in the GPa range for healthy mature bone tissue [38]. In bone regeneration, the porous structure of the scaffold material is of the utmost importance and needs to allow for cell reorganization and vascularization. Therefore, porosity, but also pore size and interconnectivity, need to be optimized without compromising the mechanical requirements. A pore size of around 100 µm is reported to facilitate cell migration, thus promoting the early stages of bone formation; this is characterized by cell recruitment, subsequent proliferation, differentiation and ECM formation. However, vascularization is of paramount importance for proper bone tissue formation, which requires larger pore sizes. Therefore, multiscale porous scaffolds that feature small and large pores might perform best in the context of tissue regeneration [39,40,41]. In addition, a high degree of pore interconnectivity is important for proper cell distribution and attachment, and for the in-growth of host blood vessels.

In summary, the key challenge in designing a scaffold used for regenerative approaches is the optimization of its microstructure, mechanical characteristics, degradability and material composition, in order to ensure a high degree of osteoinductivity, osteoconductivity and osseointegration (Figure 1). To achieve these objectives, a variety of materials and manufacturing techniques can be employed.

#### 3.1.2. Materials

Different natural and synthetic materials can be employed for creating biomimetic scaffolds, which can be used for the treatment of segmental long bone defects. Within the following section, we want to provide an overview of the most common materials employed, and also mention their main advantages and limitations. A detailed description of their application in large-animal models or clinical studies is the subject of later sections.

##### Autologous and Allogenic Bone Grafts

Given the complex interplay of the above-mentioned parameters, their optimization, in order to achieve satisfying tissue regeneration, is challenging. Therefore, the best material for regenerative approaches in bone is still represented by the bone matrix. As a result, the use of autologous bone grafts is still considered the gold standard in clinical practice [42], as it leads to excellent outcomes. Yet, autologous bone is limited in availability and the harvesting procedure introduces the need of an additional surgical site, along with the risk of donor site morbidity and infection [43,44]. Alternatively, allografts that are harvested from a distinct donor can be used. Allografts are usually harvested from cadaveric sources or during surgeries where bone material needs to be removed; therefore, they are quite heterogeneous in quality. In addition, bone grafts of xenogenic origin might be employed for the treatment of bone defects [45,46,47].

Due to the risk of disease transmission, allografts and xenografts need to be further processed by sterilization to ensure the absence of pathogens, or by deproteinization to prevent immunogenic reactions [43]. However, these procedures influence osteoinductivity and osteoconductivity, and can also diminish the mechanical properties of the allograft [36,48]. Xenogenic grafts might also be rejected due to ethical or religious concerns, and represent the least explored source for bone grafts [49].

Because of these limitations related to the use of autologous and allogenic bone grafts, natural or synthetic materials can be employed as bone substitutes [42]. For this, the engineering of porous scaffolds that provide mechanical support and allow for cell in-growth has been explored extensively for the treatment of large segmental bone defects.

##### Natural Polymers

The ECM of bone is a composite material made of an organic phase that consists of Col I, providing elasticity, and an inorganic phase comprized of HA, facilitating mechanical stability. Since Col I is one of the main composites of the native bone ECM, it has been vastly explored as a scaffold material. It offers excellent biodegradability, biocompatibility and facilitates cell adhesion [50]. However, Col I alone has weak mechanical properties that limit its use in applications without load bearing. Furthermore, Col I needs to be isolated from xenogenic materials to yield sufficient quantities. The process of Col I formation in vivo involves transcription, translation, posttranslational modification, cellular excretion and extracellular assembly into larger fibrils and fibres. Purifying collagen for tissue engineering approaches requires the breakdown of these fibres, followed by extraction and purification of polypeptide chains. Using these building blocks as a scaffold material requires additional crosslinking or blending with other materials to re-polymerise them into fibres [51]. Therefore, the use of Col I as a scaffold material is associated with a number of limitations.

Xenogenic polymers, such as silk, alginate and chitosan, have also attracted considerable interest for scaffold preparation, as they possess favourable characteristics. Silk is characterized by a good wettability as it absorbs and releases H_2_O. Moreover, silk shows a high resistance to ultraviolet light and oxidation, thereby enabling easy sterilization. It is furthermore characterized by a high tensile strength [52]. In addition, silk has a good processability and availability, and has been successfully tested in the context of bone regeneration in small animal models. Yet, further studies in large animal models are needed before silk-based scaffolds may enter clinical practice [53].

Chitosan is a polysaccharide made of chitin that is found in invertebrates and mushrooms, therefore offering excellent availability. Alginate has been investigated for various applications, including regenerative approaches in bone, as it can be easily purified and processed [54]. Similar to Col I, chitosan and alginate are biocompatible and degradable, with a rather low mechanical strength; this, therefore, limits applications for the pure biopolymer [55]. Currently, the clinical application of chitosan is restricted by a low batch-to-batch reproducibility, and by the lack of studies that have investigated its application as a scaffold material [56].

##### Synthetic Polymers

Synthetic polymers offer some advantages over natural polymers, as they can be produced in high quantities with high consistency across different production lots. Synthetic polymers also allow for chemical modification. Among the vast variety of polymers, aliphatic polyesters and their copolymers have been extensively investigated for their use as scaffold materials [37]. The most commonly used polyesters are poly-(lactide-co-glycolide) (PLGA), polyglycic acid (PGA), polycaprolactone (PCL) and poly(L-lactic acid) (PLLA). These compounds are known to be biocompatible, non-immunogenic and degradable, while providing good mechanical stability initially [57]. Most aliphatic polyesters are degraded into smaller fragments and monomers that can be easily excreted by the human body. The matrices made from these polyesters follow an erosion rate of PGA > PLGA > PLLA > PCL [58]. Degradation of these polymers in vivo is accompanied by the rapid loss of mechanical stability, the possible introduction of toxic by-products, and changes in local pH values [59,60]. Therefore, their rate of degradation requires optimization by blending with other polymers. Aside from polyesters, other polymers, such as polypropylene (PP), polyethylene (PE), polyurethane (PU) and poly (methyl methacrylate) (PMMA), have been tested as scaffolding materials for bone tissue engineering [61].

##### Bioceramic and Bioglass

Orthophosphates are a common subject for investigations related to bone regeneration, due to their similarity to the inorganic crystalline phase of bone, which mainly consists of HA (Ca10(PO4)6(OH)2) and accounts for 60% of the extracellular matrix [62]. These materials are biodegradable, show promising biocompatibility, bioactivity, osteoconductivity, mechanical strength and can be sterilized without altering their properties [63]. The most commonly used calcium phosphates are HA and beta-tricalciumphosphate (beta-TCP). Pure HA allows for the fabrication of porous scaffolds, is chemically very stable, but is of low solubility [36]. It provides good osteoconductivity but low osteoinductivity. In addition, pure HA is only slowly degraded over time, thereby impeding proper bone remodeling [42]. Pure phase beta-TCP is a well-established scaffold material for bone regeneration and has found its way into routine dental applications. This material offers a wide variety regarding porosity and pore sizes, in a range of 5–500 µm, and is, similar to HA, reported to be osteoconductive but not osteoinductive [64]. Compared to HA, beta-TCP degradation is fast and usually takes 4–6 weeks. This short time frame is insufficient to provide the required mechanical support for early bone formation [65]. To unite their advantageous properties and to optimize degradation kinetics, a combination of HA and beta-TCP (biphasic calcium phosphate (BCP)) can be employed [66].

Bioactive glass (BG) is a silicate-based material that also contains calcium and phosphorous, and is available in different compositions. This class of material has a reactive surface that supports the formation of amorphous calcium phosphates or HA, and also allows for protein absorption and cell attachment. Therefore, BG easily integrates into the bone matrix, resulting in a high osteoconductivity [67]. During degradation, BG releases Na, Ca, Si and P ions, depending on its exact formulation. These ions can increase the induction of osteogenesis and angiogenesis, which is the reason why BG is considered osseoinductive upon degradation [68]. The rate of material breakdown can be controlled as a function of its exact composition, therefore enabling the optimal support of endogenous bone remodeling [69]. The mechanical properties of BG are rather unfavorable for all load-bearing applications in bone, due to its high brittleness [70].

##### Metal

Metal and metal alloys have been extensively applied in orthopedics, e.g., for the fixation of fractures and joint replacement. Common metal implants are non-degradable and are meant to completely replace defective or missing bone tissue, instead of facilitating regeneration. However, the mechanical strength and fatigue resistance of metal alloys can be adjusted [71]. Although metal materials are not favored for regenerative approaches, there are situations in which metal implants represent the only practicable option. In scenarios were regeneration is compromized, while mechanical stability is of urgent need (e.g., tumor treatment, load-bearing bone defects), porous metal implants might be the only alternative. Here, Titanium (Ti) and Tantalum (Ta) can be used to create porous implants that permit tissue in-growth [72,73,74]. However, it needs to be considered that the utilization of metal implants usually results in the generation of metallic corrosion and wear products, including the release of ions that can lead to adverse reactions in the surrounding tissue and to elevated systemic metal levels [75,76,77].

The fabrication of degradable metal implants appears also possible when choosing elements like Magnesium (Mg) or Iron (Fe). When compared to Mg, Fe degrades at a comparably slow rate [78]. Alloying these metals can aid in tweaking their properties towards optimized rates of degradation, while maintaining their advantageous mechanical properties [79].

##### Composites

The above-mentioned materials all come with distinct advantages and limitations, while none of them can be regarded as an ideal bone substitute. Thus, creating composite materials that combine the strengths of the respective materials has been explored to design advanced biomaterials for bone regeneration [80]. Since the number of material combinations that have been studied is vast, this review will mention a few composites that aim to mimic the ECM composition of bone; the natural composite is made of the polymer Col I and the crystalline calcium-phosphate HA. Therefore, the combination of a polymer and a calcium phosphate has been explored in depth, and has also led to a number of commercially available ceramic reinforced composite materials [81,82]. In particular, the combination of collagen and HA has been investigated due to its similarity to the native bone matrix. While the combination of both materials leads to improved bioactivity and osteoinductivity, the mechanical properties of these composites were still not sufficient for load-bearing applications [83,84,85]. Since BG on its own has many advantageous properties, this material has also been vastly explored for the creation of composites that can be used in bone regeneration [86,87]. While, in theory, composites offer the creation of true biomimetic scaffold materials, recreating mature bone in terms of mechanical and physicochemical properties has not been achieved yet. In summary, none of the available materials can be regarded as an ideal bone substitute, and they rather have specific advantages and limitations that should be used according to the clinical problem (e.g., load-bearing or non-load-bearing defect regeneration). The combination of two or more materials has great potential to create materials that mimic bone as a composite material more closely. Yet, the optimal material combination for bone regeneration is still to be found.

#### 3.1.3. Manufacturing Methods

A number of different methods are available for the fabrication of porous scaffolds. Traditional methods lack the consistency and reproducibility of results, and only offer limited control over spatial parameters; meanwhile, additive manufacturing has introduced the possibility of producing 3D scaffolds with a high degree of control, regarding porosity and overall shape. Among classic approaches, the combination of solvent casting with particulate leaching can be used to produce porous polymer scaffolds (Figure 2A). The polymer is brought into solution in its organic solvent and mixed with the particulate, often a salt, which is insoluble in the solvent. The solvent then evaporates and the mixture of the polymer and particulate is submerged in water to dissolve the salt particles, thus leaving a porous structure [88]. While this process is rather cost-effective, it lacks precise control over pore size and interconnectivity [89].

The technique of gas foaming involves the dispersion of gas bubbles throughout a polymer at high pressures (Figure 2B). Decreasing the pressure to ambient levels will then lead to a nucleation of gas bubbles. The gas subsequently starts to diffuse out of the polymer, thereby creating a porous structure [90]. This approach is rather fast and comparable cheap, yet, similar to solvent casting, the created pores lack interconnectivity and it is difficult to control the pore size.

In freeze drying, an emulsion of a polymer/solvent mixture and water is cooled down to facilitate the separation of solvent ice crystals and the surrounding polymer (Figure 2C). The subsequent application of a vacuum facilitates the sublimation of the solvent and water, thereby creating pores [57]. Since this technique works at low temperatures, the addition of biomolecules or the use of natural scaffold polymers is possible [91,92]. Yet, the process is afflicted with the formation of pores with irregular sizes and the possibility of solvent residuals that remain within the scaffold [68].

The procedure of thermally induced phase separation (TIPS) describes the unmixing of a homogenous polymer/solvent solution through changes in the thermal energy (Figure 2D). After phase separation, the solvent is removed by subsequent freeze drying [93]. TIPS enables good control over pore geometry but can only be applied for polymers with low melting temperatures [94].

In the process of electrospinning, fibers in the micro and nanometer range are created by a high voltage electric field that draws charged threads of polymers from a capillary tube onto collector plates (Figure 2E) [95]. Electrospinning is simple and cost-effective, but the resulting fiber-based scaffolds, however, lack mechanical stability [96].

**Figure 2 biomedicines-11-00325-f002:**
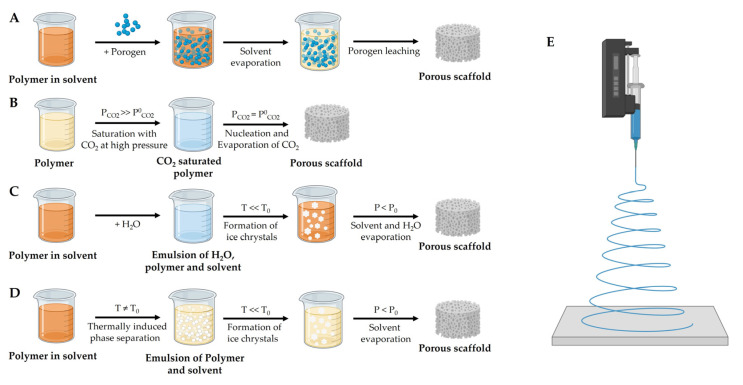
Different methods for the manufacture of porous scaffolds. (**A**) A polymer is brought into solution in its solvent form and mixed with porogen particles. After solvent evaporation, the porogen is leached out, leaving a porous structure behind. (**B**) For gas foaming, a polymer is saturated with an inert gas under high pressure. Releasing the pressure to ambient levels results in the nucleation of gas bubbles, followed by evaporation of the gas, resulting in a porous scaffold. (**C**) In freeze drying, an emulsion of a given polymer in its solvent and water is created. After the temperature is lowered, ice crystals are formed within the polymer, the subsequent application of a vacuum facilitates the solvent and water evaporation thereby creates porous structures within the scaffold. (**D**) in TIPS, a phase separation between the polymer and solvent is induced by changes in the thermal energy. Adjacent freeze drying facilitates solvent evaporation. (**E**) Electrospinning utilizes a high-voltage field to draw a polymer filament onto a collector plate. Figure is partly adapted from [96]. Figure contains illustrations and icons created with BioRender.com.

Additive manufacturing describes the method of the automated layer-by-layer deposition of a material to create a 3D object. This process is controlled by a computer-aided design (CAD) file that contains the volumetric information of the printable object (Figure 3A). In recent years, this technique has become increasingly accessible and has been vastly explored for the fabrication of scaffolds for tissue engineering [97,98].

In stereolithography (SLA), photopolymerizable liquid monomers are crosslinked and hardened by UV light to form 3D objects made of the respective polymers (Figure 3B) [99]. This method offers a high resolution and print accuracy, yet the choice of materials is limited. Furthermore, the use of photocrosslinkable monomers means that the used resins are often toxic, thus necessitating the use of protective gear. Printed objects need postprocessing to remove residual monomers and additional crosslinking under UV light [100,101].

The process of selective laser sintering (SLS) is based on the fusion of polymer or metal particles by a laser, while new layers of particle powder are added by a roller into the print bed (Figure 3C). This method allows for the processing of a wide range of materials and good control over mechanical properties. The fusion of particles, however, might introduce unwanted porosities, thereby creating heterogeneities in mechanical properties and leading to the thermal degradation of the sintered material [102,103].

The technique of fused deposition modelling (FDM) uses a temperature-controlled extruder for the short term melting of polymer filaments (Figure 3D). FDM permits the layer-by-layer deposition of materials with moderate resolution at relatively low costs. The range of suitable materials is comparably wide and some printers offer simultaneous use of multiple materials. In contrast to SLA or SLS, the formation of larger cavities might require the use of a sacrificial ink, thereby limiting the geometry of possible designs [104].

**Figure 3 biomedicines-11-00325-f003:**
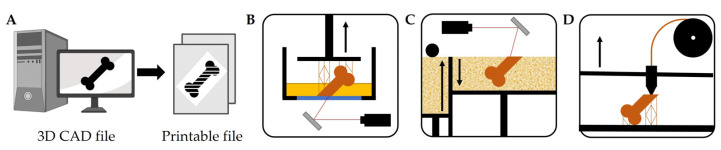
The three main methods for 3D printing. (**A**) Segmentation of a 3D CAD file into a format that allows a layer-by-layer reconstruction of the original volumetric body by 3D printing. (**B**) In stereolithography, the printing platform is submerged in a liquid monomer solution. The monomers are crosslinked into a polymer and hardened layer-by-layer by UV light, provided by a laser or LEDs. (**C**) In selective laser sintering, particles provided as a powder are fused layer-by-layer by laser light and a new powder is provided from a separated reservoir via a roller. (**D**) In fused deposition modelling, a layer-by-layer deposition of melted polymer filament is facilitated by using a thermal print head. Figure is partly adapted from [105]. Figure contains illustrations and icons created with BioRender.com (accessed on 8 December 2022).

Bioprinting describes additive manufacturing of tissue constructs such as bone substitutes that incorporate cells or bioactive molecules into the printing process [106]. This approach requires the simultaneous processing of scaffold material and biological components in the form of a printable bioink [107]. While conventional additive manufacturing is based on photopolymerization and heat assisted material deposition or sintering, bioprinting is limited to temperatures that allow for cell survival and that prevent the denaturing of added growth factors. In addition, shear stress needs to be kept low to prevent cell death, thereby limiting extrusion speed and nozzle sizes. Most bioinks are, therefore, based on low-viscosity natural polymer hydrogels that ensure good cell viability and function, but exhibit low mechanical strength, limiting their range of application [49]. Photocrosslinkable bioinks allow for polymerization by using UV-light, and thus offer the application of SLA approaches that offer higher print resolution without the generation of shear stresses [108]. Yet, it needs to be considered that such formulations might be based on rather toxic monomers. In addition, the application of UV light might introduce DNA damage to cells [109]. Despite these technical challenges, bioprinting offers undeniable advantages over conventional manufacturing methods. These include the possibility to create cell-laden, defect-fitting scaffolds, and the addition of signaling molecules, such as angiogenic factors, to enhance vascularization [110]. In addition, bioprinting offers precise control over spatial geometries, thereby allowing for the recreation of tissue specific structures, such as the trabecular bone. Therefore, bioprinting approaches have generated considerable interest in the context of bone tissue engineering [111,112].

In a recent study, it was demonstrated that the in situ printing and UV-hardening of a bioink is possible in an animal model, demonstrating that this approach might be applicable for the treatment of large segmental defects [113].

In conclusion, the manufacturing method applied for scaffold fabrication needs to be selected according to the properties of the material employed and the desired scaffold design parameters.

#### 3.1.4. Cellular Components

In bone regeneration, scaffold materials are meant to provide structural guidance while promoting the adhesion, growth and differentiation of endogenous bone-forming precursor cells. However, the sufficient in-growth of tissue resident cells might be hampered by the sheer size of the defect or by prevalent pathologies. Here, the augmentation of a scaffold with bone forming cells or their progenitors can help to facilitate proper tissue regeneration [114]. The possible sources for osteogenic cells include embryonic stem cells (ESCs), which can give rise to cells from all germ layers. Yet, the use of ESCs in bone regeneration is limited due to ethical considerations and availability, relating to their origin and the corresponding regulatory limitations [115]. In addition, the use of ESCs might introduce the risk of teratoma formation [36]. With the introduction of induced pluripotent stem cells (iPSCs), a pluripotent cell source without the drawbacks regarding ethics or accessibility was made available. Despite their great potential, iPSC usage still harbours the risk of teratoma formation as a result of insufficient reprogramming into the desired phenotype. Therefore, iPSCs have to be further investigated prior to clinical application in bone tissue regeneration [116].

To date, the most promising approach is the transplantation of auto or allogenic osteogenic precursor cells, such as MSCs. These cells have a long history and track record in regard to musculoskeletal regenerative approaches, due to their advantageous properties and relative ease of isolation and expansion [117]. MSCs possess a multilineage differentiation potential and can acquire the phenotype of musculoskeletal cells, such as osteoblasts and chondrocytes [118]. It is also well established that MSCs constantly replenish the rather short-lived osteoblasts in vivo [119]. In addition, undifferentiated MSCs lend themselves for allotransplantation, since they express a major histocompatibility complex (MHC) class I, but not MHC II molecules [120,121]. In addition, these cells possess immunomodulatory and immunosuppressive capabilities, thus having a low tendency to elicit an immune response by the host [122,123,124,125]. In terms of availability, MSCs can be isolated from a number of different tissues, while bone marrow, adipose tissue and umbilical cord are the most attractive sources for regenerative approaches [36]. In summary, MSCs are the most promising cell type for bone tissue regeneration and have, therefore, been combined with polymers or calcium phosphate scaffolds in a multiplicity of studies [27,126]. A silk-HA composite scaffold with good mechanical properties was successfully seeded with human MSCs that exhibited good survival and proliferation [127]. MSCs derived from human adipogenic tissue (AT-MSCs) were seeded onto scaffolds made of Col I, Col I/beta-TCP and Col I/HA/beta-TCP, leading to the activation of different osteogenic pathways as a function of the exact material combination utilized [128]. MSCs, of human umbilical cord origin (hUCMSCs), were successfully used to augment a Col I calcium phosphate cement scaffold [129]. Culture under perfusion can help to further promote osteogenesis prior to implantation, as shown for human MSCs (hMSCs) in combination with a beta-TCP-based scaffold [130]. The co-culture of MSCs with human umbilical vein endothelial cells (HUVECS) on a beta-TCP/Calcium silicate scaffold facilitated not only osteogenesis, but also angiogenesis [131]. While all of these examples demonstrate the general feasibility of seeding scaffolds with cells, examples for the implementation of such an approach in a clinical setting remain scarce. A microporous HA scaffold was loaded with autologous MSCs to treat a single patient with a large bone defect. Although anecdotal, the practicability of this approach was clearly demonstrated, while an improvement in the defect repair was also reported [132]. The successful combination of a polymeric carrier material and osteogenic cells for the augmentation of the maxillary sinus was reported in a larger cohort study [133].

Taken together, a great number of studies provide evidence that bone defect augmentation with cell seeded scaffolds is possible; however, the approach faces limitations associated with cell sources, manipulation and expansion in vitro, since these aspects drive the associated costs and play a pivotal role in the process of regulatory approval [134,135,136].

#### 3.1.5. Growth Factors

Another approach to improve scaffold-based tissue regeneration relies on the addition of growth factors [137]. These soluble signaling molecules are locally secreted by cells and diffuse over small distances within tissues in vivo [138]. Growth factors are usually characterized by a low stability, resulting in a rather short duration of action upon release, while their overexpression and subsequent accumulation lead to unwanted side effects.

Among other growth factors, BMPs and VEGF play an important role in bone regeneration [139]. BMPs belong to the TGF superfamily and were first recognized for their ability to induce ectopic bone formation [140]. In vivo, the expression of different BMPs govern bone formation and skeletal growth [141]. Their temporal and spatial release needs to be well-orchestrated in the light of regenerative approaches to avoid adverse effects, such as an increase in bone resorption, inappropriate adipogenesis, and unwanted ectopic bone formation or tumor formation [142,143]. In addition, it was demonstrated that BMP predominately induces intramembranous rather than endochondral ossification, depending on dose applied [144].

The growth factor VEGF is involved in angiogenesis throughout all tissues within the body, including bone, where a proper vascularization is of the utmost importance for regeneration. The overexpression of VEGF on a local or systemic level might cause hypotension or a decreased cardiac stroke volume [145]. Therefore, the application of growth factors requires precise control over release kinetics in terms of dose, localization and time frame of release [146]. In this context, surface representation or the degradation-mediated controlled release of growth factors from scaffolds are regarded the most promising approaches for bone regeneration [137]. For example, the surface coating of PLGA scaffolds with BMP-2 and PDGF, and their subsequent controlled release, was demonstrated to have a positive effect on bone regeneration in a critical-size defect model in rats [147]. The dual release of BMP-2 and VEGF, upon the degradation of gelatin microspheres embedded in a porous poly (propylene fumarate) scaffold, was used to successfully enhance bone regeneration in small animals [148]. Since the process of bone regeneration is of sequential nature, different cells and growth factors act at different stages of tissue formation. Therefore, the sequential release of VEGF and BMP-2 from PLGA microspheres was investigated in an animal model and shown to be beneficial for guided bone regeneration [149].

In conclusion, the controlled release of one or more growth factors from a scaffold material is feasible and can help to improve bone regeneration.

### 3.2. Vascularization Strategies

Bone is a highly vascularized system and receives up to 10% of the cardiac output. Functional revascularization of the traumatized tissue is crucial for fracture repair, especially in long bone defects, in order to supply oxygen and nutrients, to remove debris and to allow for the recruitment of circulating cells. Bone-forming osteoblasts require the proximity to a capillary of less than 300 µm, while chondrogenesis is driven by a reduced oxygen and nutrient supply [150]. During endochondral bone repair, the lack of oxygen in the cartilage template regulates the chondrocyte metabolism and induces the expression of VEGF [151]. Since hypertrophic chondrocytes produce significant amounts of VEGF, they are considered key drivers in vascularization, while osteoblasts co-mobilize with new blood vessels to induce bone healing [152]. Thus, a compromised vascular supply and insufficient revascularization are primary clinical problems in long bone defects, and a low peripheral vascular supply with an obliterated vascular bed are more prone to resulting in non-union formation.

The biggest challenge in scaffold-guided bone regeneration is to ensure blood supply in order to bypass a potential lack of neovascularization, subsequent ischemia, and necrosis. Strategies to overcome the challenge of guiding new vessel formation are based on accelerating angiogenesis into the scaffold (e.g., adapted scaffold design, angiogenic growth factor delivery) or using the pre-existing vasculature to be connected for sufficient blood supply (e.g., axial vascularization or pre-vascularized scaffolds) (Figure 4). As mentioned before, scaffold design parameters, such as porosity, pore size and pore interconnectivity, are crucial to allow the penetration of pre-existing vessels and vasculature-forming cells. It was demonstrated that large-pored Poly (ether ester) block-copolymer scaffolds (>250 µm) resulted in a higher vascular density when compared to scaffolds with smaller pore diameters after subcutaneous transplantation in balb/c mice [153]. This is in line with other studies reporting the superiority of a larger pore size to support vascularization, e.g., using Ti6Al4V scaffolds [154] or beta-TCP [155] (both pore sizes >500 µm). In general, it has been suggested that interconnected larger pores (>600 µm), as well as a porosity of (>70%), are favorable to drive blood vessel in-growth and, therefore, bone regeneration [156].

Several growth factors are of great interest for vascularized tissue engineering constructs. VEGF, PDGF and FGF have been comprehensively studied to attract endothelial progenitors for enhanced angiogenesis in the implanted scaffold. However, it is especially crucial to determine the optimal time point and rate of growth factor release that can be modulated by different delivery strategies (e.g., physical direct embedding in hydrogels, encapsulation in microparticles or chemical immobilization to control release by degradation of vehicle) [157]. VEGF, for example, has a narrow therapeutic window and a short half-life, with potential toxic effects resulting in altered, non-functional vessel formation upon higher or long-term dosing [158,159]. These potential side effects and complications could be the reason why no clinical trials have been published so far [160], despite wide evidence in the literature for the ability of scaffold-combined VEGF release to successfully support bone regeneration in cranial or calvaria defects [161,162,163]. Additionally, a controlled release of VEGF can be achieved by modifying the scaffold with heparin [164,165]. Further studies explore the potential synergistic effect of VEGF combined with, e.g., BMP-2 [148,166,167,168,169] or BMP-6 [170], in order to initiate both pro-angiogenic and pro-osteogenic cues. A comparable biphasic scaffold approach was proposed by combining a calcium phosphate cement paste (osteoconductivity) with an alignate/gellan gum hydrogel paste loaded with VEGF (vascularization), in order to improve segmental bone defect repair [171]. In addition, it was reported that BMP-2 alone activates paracrine signaling in osteoprogenitors to induce neovascularization [172,173]. To overcome the short half-life of promising growth factors, such as PDGF, novel approaches explore ways to, for example, combine bio-mineral PDGF-coated fibers with hADSCs in spheroids [174], or to genetically modify MSCs to overexpress PDGF-BB [175]. The indirect stimulation of VEGF expression by local resident cells can be additionally achieved by, e.g., the delivery of therapeutics targeting the hypoxia-inducible factor (HIF) pathway. Among other therapeutic agents, Deferoxamine (DFO) was shown to be promising when delivered with or without a scaffold in bone defect sites, as it especially accelerates angiogenesis and revascularization [176,177,178,179,180]. The secretome of hypoxia-conditioned MSCs was incorporated in a scaffold to direct the MSC migration and tube formation of HUVECs in vitro [181].

Further strategies, aimed at the revascularization of implanted scaffolds for bone tissue engineering, employ in vitro pre-vascularization approaches by embedding HUVECs or endothelial progenitors into the scaffold to form a vascular network, prior to implantation. Roux et al. reported that HUVECs encapsulated in fibrin scaffolds formed extensive vascular networks in vitro, which were maintained and anastomosed with the host tissue upon implantation into calvarial defects in nude rats [182]. Further studies investigated the possibility of combining HUVECs with, for example, osteoblasts [183] or MSCs [184,185]. It was reported that cell–cell communication between HUVES and MSCs supported the stabilization and maturation of the newly formed vasculature in vitro and in vivo, which is in line with other studies [186,187].

Bioprinting methods allow the control of the spatial distribution of cells in the scaffold, which is a great advantage when, e.g., co-embedding HUVECs and osteoprogenitor cells. This strategy has been successfully investigated by combining adipose-derived stromal cells (ASCs) and HUVECs [188], or by using MSCs together with HUVECs [186]; this shows promising results towards the production of pre-vascularized artificial bone tissue. Since these methods normally result in randomly organized networks, recent developments in biomanufacturing approaches have paved the way towards the fabrication of pre-vascularized scaffolds, comprising engineered functional capillaries to potentially create anastomosis with the pre-existing host vasculature upon implantation [157]. Recent approaches, therefore, explore the technical feasibility to fabricate microchannels, which can be further functionalized by seeding endothelial cells [189,190,191,192,193].

Despite the tremendous development and the efficient use of emerging technologies in biomanufacturing, most of the previously described vascularization strategies, including growth factor application or functional approaches towards scaffold pre-vascularization, that rely on cell embedding or microchannel fabrication, have not yet been translated to the clinic.

Vascularized bone transfer has been an emerging surgical reconstruction technique, based on the growing knowledge about the complex vascularity and the anatomy of the blood supply in bone [194]. The concept of following or recreating the natural vasculature has been transferred by using an axial vascularization approach to provide blood supply and integration for newly implanted scaffolds. Consequently, the bone-engineered scaffold is first implanted into a region with a sufficient vascular supply that allows anastomosis with the pre-existing vasculature [195,196]. Ultimately, the arteriovenous system is used as a pattern to direct scaffold vascularization along its longitudinal axis [156]. Extrinsic and intrinsic approaches have been investigated, differing in the spatial alignment of the vessels to the scaffold. However, the intrinsic variant has proven to be advantageous, providing the center of the scaffold with a blood flow that subsequently leads to more uniform tissue remodeling [197,198]. Alternatively, flap-based methods employ the established vascular network of another tissue, such as periosteum, muscle, or omentum, for axial scaffold vascularization [156]. In a second step, after the formation of a microvascular network, the vascularized scaffold is harvested and implanted into the bone defect with microsurgical anastomosis, ensuring the immediate perfusion. This method is also referred to as an in vivo bioreactor [156,199] or prefabrication [200]. In this regard, the successful implementation of the matching axial vascularization approach in four clinical cases has been recently reported; this will pave the way for further future clinical trials [201].

## 4. In Vivo Application of Scaffolds in Bone Regeneration

### 4.1. Large Animal Models

Large animal models are advantageous for the effective testing of new scaffolds, biomaterials, or implants for orthopedic applications. The comparable bone size, high body weight and similar biomechanical properties (weightbearing) allow for a human-relevant testing scenario to promote clinical translation. The advantages and translational potential of large animal models are not discussed here and we recommend the following reviews for further reading [202,203,204]. However, approaches that show success in large animal studies tend to be translated into clinical application. Table 1 lists the selected preclinical studies of large animal models, testing different scaffold approaches for full-thickness, large-bone defect healing.

In this regard, composite scaffolds are of particular interest and have been recently tested in large animals; their composites include, but are not limited to, titanium/polyamide [221], beta-TCP/biopolymer [222], polyurethane/HA/decellularized bone particle [212], and calcium phosphate/alginate [223] or poly-L/D-lactide copolymer/barium titanate nanoparticle [224]. Metallic scaffolds, such as titanium-based porous [74] or mesh scaffolds [206], are preferably tested for the optimal stabilization of critical segment defects. Approaches using polymers have also been explored in large animals. For example, the combination of 3D-printed mPCL scaffolds with BMP-7 and Platelet-Rich-Plasma was tested for bridging a large volume segmental bone defect (19 cm^3^) in sheep [219]. In a 40 mm tibial defect model in sheep, an external hybrid-ring fixator was used to test hollow cylindrical ceramic-polymer composite scaffolds with or without bone grafting [205]. Other studies focused on ceramic scaffolds, such as biomorphic calcium phosphate (GreenBone™) [220] or strontium–hardystonite–gahnite [210]. The element strontium has been tested in several studies as a bioactive coating or for biomaterial supplementation, as it promotes bone formation [225,226,227,228]. Since 3D printing techniques have been tremendously improved in recent years and also allow for the processing of ceramic materials, several new structurally optimized 3D printed scaffolds have been tested in large animal models [213,229,230,231]. Some of the recent approaches also explore the potential feasibility of novel biomaterials, such as nacre [232,233], keratin [234] or the integration of plant-derived soybean peroxidase [235].

In addition, the effective and safe delivery of growth factors, such as BMP-2 [211,216,217,236] and VEGF [237,238], has been tested over the last 4 years in large animal models. For example, a rather complex strategy, using an in vivo bioreactor approach to pre-vascularize a 3D-printed BMP-2-loaded scaffold before implantation into a 5 cm segmental defect in the ovine tibia, was tested and found to be feasible [216]. In another study, a 3D printed polymer/ceramic composite scaffold was successfully combined with a rhBMP-2-loaded collagen sponge to bridge a critical size defect (5 cm) in a sheep metatarsus fracture model (intramedullary nail stabilization) [217]. Two different groups studied PCL/beta-TCP scaffolds or porous beads as carrier materials for the release of BMP-2 in dogs [211,236]. Cell-based approaches have been increasingly tested in vivo in large animal models, comprising cord blood cells [209], osteoprogenitors [218], mesenchymal stromal cells [207,214,239,240], periosteum-derived stem cells [215] but also blood [208] or platelet-rich fibrin exudate [241]. In summary, promising results were achieved by using composite scaffolds in animal models. The addition of osteogenic and endothelial cells greatly enhanced the successful integration of the graft. The supplementation of the scaffolds with growth factors or with other osteogenic stimuli, such as strontium, also promoted implant integration and large defect repair. These concepts have, therefore, been proven successful, meaning that their further translation into clinical routine seems feasible.

### 4.2. Clinical Application

One focus of musculoskeletal research efforts in the past two decades has been on the development of bone graft substitutes that provide sufficient biomechanical support and allow reliable scaffold fixation at the site of transplantation. However, only a few novelties managed their way to clinical translation. The revised European medical devices regulation (EU 2017/745), following the safety by design paradigm, demands a thorough preclinical assessment within internationally approved standards (ISO, ASTM) to demonstrate a product’s safety and efficacy [61]. The process of conformity assessment ensures that novel products meet legal and safety requirements, and perform as intended. In the case of SmartBone^®^ (IBI), a bovine mineral bone matrix was combined with bioresorbable polymers (poly(lactic-co-caprolactone)) and arginine, glycine and aspartic acid (RGD) peptide-presenting collagen fragments. After thorough preclinical evaluation in vitro, in animal models [242] and in clinical trials [243,244], the bone substitute material received a CE marking, in accordance with the 93/42/EEC medical device directive, and is classified as a class III medical device.

Other bone substitute materials that underwent successful translation include, for example, growth-factor-activated products; the majority of these consist of allografts (OsteoAMP^®^, Bioventus Surgical) containing a variety of different growth factors or collagen/beta-TCP scaffolds combined with PDGF (Augment^®^ bone graft, Wright Medical). Clinical trials utilizing these materials have proven their ability to promote bone union in patients undergoing lumbar body fusion, as well as ankle or hindfoot arthrodesis [245,246,247]. The growth factors incorporated in these scaffold materials are, however, associated with a short-term activity, limiting clinical efficacy.

To circumvent these activity-related disadvantages, the release of plasmid DNA from scaffolds encoding for specific growth factors, such as VEGF, was introduced to facilitate the local production of reparative factors, driving regeneration. The feasibility of this approach was demonstrated by the reconstruction of maxillofacial bone defects utilizing a collagen-HA or octacalcium-phosphate bone substitute, equipped with plasmid DNA encoding for VEGF-A (Histograft^®^, Histograft LLC) [248,249].

The application of mesenchymal progenitor cells, in order to stimulate bone healing, is appealing and has been subject to extensive research. Surprisingly, these cell-based treatment strategies have contributed only marginally to current clinical practice. Injections of autologous bone marrow aspirates were first applied in 1991 to aid bone healing in tibial non-unions [250]; bone marrow, in combination with a freeze-dried corticocancellous allograft, achieved similar results when performing posterior spinal fusion in case of progressive adolescent idiopathic scoliosis [251]. Similar outcomes were observed in patients undergoing posterolateral spinal fusion after the application of a composite collagen-HA sponge (Healos^®^, DePuy) [252] or porous beta-TCP granules [253] after incubation with autologous bone marrow.

Percutaneous injections of allogeneic demineralized bone matrix and autogenous bone marrow were furthermore used to treat active unicameral bone cysts (n = 23) after trephination [254].

To allow improved standardization regarding administered cell types and numbers, and to allow cell enrichment and pre-differentiation with the aim of increasing the therapeutic potential, the ex vivo expansion of mesenchymal autologous cells appears attractive.

Consequently, Sandor et al. administered in vitro expanded adipose-derived autologous stem cells [255] to 13 cases of cranio-maxillofacial hard-tissue defects at four anatomically different sites. The cells were seeded onto BG or beta-TCP in combination with, or without, recombinant human BMP-2 [255], achieving the successful integration of the construct to the adjacent bone in 10 cases. Vasyliev et al. report on the treatment of 47 patients suffering from combat related osseous gunshot injuries, showing no sufficient healing after previous surgical treatment. Depending on defect size, which varied between 10 and 180 cm^3^, allogeneic bone grafts were used as a guiding scaffold in form of block, chips or in combination. These carriers were seeded with a 2–5 × 10^6^ cells/cm^3^ scaffold, directly utilizing a fibrin-based hydrogel. Cells consisted of BM-MSCs cells only in the case of defects smaller than 5 cm in length or in combination with periosteal progenitor cells at a ratio of 3:1 between 5 and 7 cm length; this is together with endothelial progenitor cells (ratio 3:1:1), in defects exceeding 7 cm in length. These constructs were transplanted after in vitro incubation for 5–14 days. Notably, radiographic controls determined healing defects in 90 percent of the treated defects within 4–6 months [256].

Another recently initiated EU-funded clinical trial (Maxibone) evaluates the use of culture-expanded autologous bone marrow stem cells, together with calcium phosphate ceramics, to facilitate personalized maxillary bone regeneration.

While all these applications demonstrate successful regeneration in a variety of bone defects, clinical trials reporting on approaches to regenerate long bone defects remain scarce. Reports in the literature are limited to case reports with a small patient cohort. For example, Laubach et al. recently reported four cases, treated utilizing methods of patient-specific scaffold-guided bone regeneration. The concept, based on the use of 3D printed composited scaffolds (PCL/beta-TCP), was previously well-characterized in large animal models of segmental defects [257,258,259,260]. The scaffolds were combined with an autologous bone graft, harvested with a Reamer–Irrigator–Aspirator System (RIA^®^, Synthes) [261].

A web-based search of the clinicaltrials.gov database, utilizing the search term “bone loss”, led to 912 search results, and a consecutive search using “bone” and “regeneration” resulted in 327 items. Thirteen of the listed trials actually related to the reconstruction of long bone defects. They are summarized in Table 2.

## 5. Future Perspectives and Concluding Remarks

Research in the field of tissue engineering, particularly in regard to large segmental defect bone regeneration, has considerably expanded in recent decades. However, only a few concepts have managed its successful translation to routine clinical application. This may be attributed to the well-known dissociation of basic research on the bench level from clinical treatment at the bedside. To resolve this translational “valley of death”, it is of the utmost importance that scientific experimentation and clinical application find a solid base in evidence-based preclinical data, investigating practicable, translatable approaches.

Despite promising clinical results after the transplantation of scaffolds, osteogenic cells and/or growth factors, the question of scaffold micro- and nano-topography, cell source, optimal cell number, growth factor dosage and mode of application, and their exact mechanisms of action, remain unclear. Concept modifications that aim to reduce local inflammation processes, increase local vascularization and provide antibacterial activity, along with dynamization-dependent, phase-bonded mechanobiological stimuli, might aid to further improve bone healing in complex cases. Given these knowledge gaps and the technology’s rather slow advance in terms of translation, it is unlikely that a scaffold-based approach, able to serve all clinical demands, will be found soon. Rather, the choice of scaffold material, manufacturing process, the addition of cells and the incorporation of growth factors are parameters that need to be chosen and balanced against each other, according to the treatment requirements. Novel concepts, such as bioprinting, are greatly promising as they allow for 3D fitting of the graft to the actual defect and the simultaneous deposition of different materials, cells and growth factors. Despite this promise, bioprinting is still a rather novel concept that needs further research, especially into the formulation of adequate bioinks. Among the vast variety of possible biomaterials applicable for classic and novel manufacturing methods, only a few formulations will pass the translational gap and find their way into clinical routine. This is especially true for formulations that include cellular components or growth factors. Here, availability, ethical considerations and regulatory requirements need to be balanced against the materials’ performance as a biomimetic bone substitute. Therefore, it is likely that well-characterized materials, which have already found their way into clinical application, such as in bioceramics, will be further functionalized, used in composites or in bioink formulations to improve the treatment of segmental large bone defects.

## Figures and Tables

**Figure 1 biomedicines-11-00325-f001:**
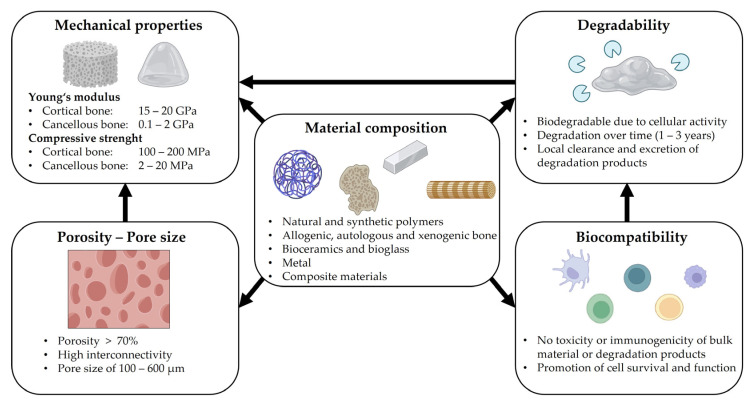
Parameters that need consideration in scaffold design and their interplay. Figure contains illustrations and icons created with BioRender.com (accessed on 8 December 2022).

**Figure 4 biomedicines-11-00325-f004:**
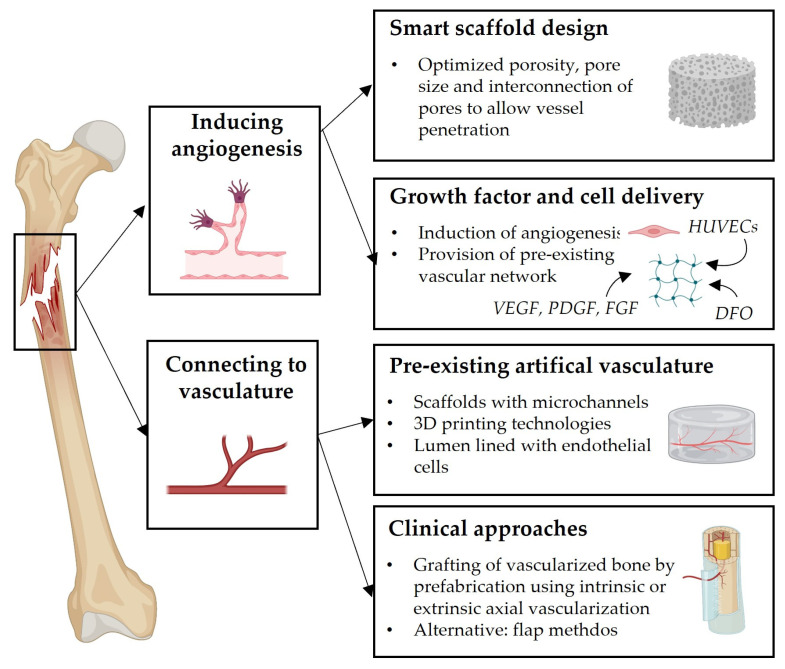
Overview on current vascularization strategies for scaffold guided bone regeneration. Figure contains illustrations and icons created with BioRender.com (accessed on 8 December 2022).

**Table 1 biomedicines-11-00325-t001:** Selected studies testing new scaffold-based approaches for full-thickness long-bone defects in large animal models between 2018 and 2022 (exemplary overview; no systematic literature analysis).

Reference	Animal Model & Defect Characteristics	Scaffold
Pobloth et al., 2017/2018 [205]	Sheep; 4 cm segmental tibial defect	Cylindrical ceramic-polymer composite scaffold (beta-TCP, -TCP, chronOS/poly-lactid co-ε-caprolactone)
Pobloth et al., 2018 [206]	Sheep; 4 cm segmental tibial defect	Titanium-mesh scaffold
Szivek et al., 2018 [207]	Sheep; 4.2 cm segmental femoral defect	Mesenchymal stem cell seeded, biomimetic 3D printed scaffold
Balaguer et al., 2018 [208]	Dog; 2 cm segmental ulna defect	Biphasic calcium phosphate microparticles in autologous blood clot
Herten et al., 2019 [209]	Sheep; 2 cm segmental tibial defect	Autologous umbilical cord blood cells on HA scaffold
Li et al., 2019 [210]	Sheep; 3 cm segmental tibial defect	Multicomponent bioceramic scaffold (strontium—hardystonite—gahnite)
Hong et al., 2020 [211]	Dog; 1.1 cm segmental femoral defect	Leaf-stacked PCL beads loaded with BMP-2
Grzeskowiak et al., 2021 [212]	Horse; 2 cm segmental defect in metacarpal bone (MCIV)	Composite scaffold consisting of polyurethane, HA and decellularized bone particles
Vidal et al., 2020 [213]	Sheep; 3.5 cm segmental metatarsal defect	A 3D printed calcium phosphate scaffold (MimetikOss 3D) with axial vascular pedicle
Crovace et al., 2020 [74]	Sheep; 5 cm segmental tibial defect	Biomimetic porous titanium (Ti_6_Al_4_V ELI) scaffold
Black et al., 2020 [214]	Sheep; 3 cm segmental tibial defect	PCL scaffold with bovine extracellular matrix hydrogel seeded with Stro-4+-enriched bone marrow mesenchymal stromal cells
Lammens, Maréchal et al., 2020 [215]	Sheep; 3 cm or 4.5 cm segmental tibial defect	Periosteum-derived stem cells on ceramic dicalciumphosphate scaffold (CopiOs) with BMP-6 or BMP-2
Yang et al., 2021 [216]	Sheep; 5 cm segmental tibial defect	Prevascularized bone graft and BMP-2 loaded 3D printed scaffold
Yang et al., 2021 [217]	Sheep; 5 cm segmental metatarsal defect	A 3D printed PCL/beta-TCP composite scaffold with rhBMP-2-eluting collagen sponge
Bajuri et al., 2021 [218]	Sheep; 3 cm segmental tibial defect	Tissue-engineered HA bone scaffold containing BMSC-derived osteoprogenitor cells
Henkel, Medeiros Savi et al., 2021 [219]	Sheep; 6 cm segmental tibial defect	Composite scaffold (PCL/beta-TCP) with platelet rich plasma (PRP) and BMP-7
Kon et al., 2021 [220]	Sheep; 2 cm segmental metatarsal defect	GreenBone—biomorphic, hierarchically structured apatitic scaffold

**Table 2 biomedicines-11-00325-t002:** Clinical trials investigating the reconstruction of long bone defects.

Title	Registration Number	Status	Phase	Condition	Intervention
Calcium Sulfate Spacer in Open Tibia Fractures with Sub-segmental Bone Loss to Decrease the Need for Secondary Surgery	NCT03042546	Terminated prematurely	4	Open tíbia fracture with sub-segmental bone loss	Calcium Sulfate PMMA No treatment
An International, Multicenter, Prospective Registry to Investigate Treatment Options and Their Outcomes on Post-traumatic Long Bones Defects	NCT04112992	Not yet recruiting	n.a.	Traumatic bone defect in any long bone	Any treatment that is used for a defect of any long bone
The Effectiveness in the Treatment of Long Bone Defect in Adults Using 3D-printed Titanium Alloy Implant	NCT04449211	recruiting	1	Bone defect greater than 5 cm due to trauma or tumor resection	Long bone defect reconstruction with 3D-printed customised Titanium alloy implant
Clinical Trial of Freeze-dried Bovine HA/Secretome Composite Application for the Management of Long Bone Defects in the Lower Extremities	NCT04980261	recruiting	n.a.	Bone defects (<5 cm) in the diaphysis of the long bones of the lower extremities due to trauma and other bone healing disorders	freeze-dried bovine HA/secretome composite vs. autograft
Treatment of Patients with Segmental Bone Tissue Defects Using Mesenchymal Stem Cells Enriched by Extracellular Vesicles	NCT05520125	Not yet recruiting	1	non-unions and segmental defects the tubular bones of the upper limbs	Mesenchymal stem cells enriched by extracellular vesicles vs. standard treatment
Bone Transport Through Induced Membrane Technique Versus Conventional Bone Transport Technique in Management of Bone Defects of Lower Limbs	NCT05631951	Active Not recruiting	n.a.	infected non-united fractures of lower limb long bones	bone transport through induced membrane vs. conventional bone transport
A Multi-centre, Open-label, Randomized, Comparative Clinical Trial of Two Doses of Bone Marrow Autologous MSC+ Biomaterial vs Iliac Crest Autologous Graft, for Bone Healing in Non-union After Long Bone Fractures	NCT03325504	Active Not recruiting	3	Traumatic isolated closed or open Gustilo I and II, IIIA and IIIB humerus, tibial or femur diaphyseal or metaphysodiaphyseal fracture with a status of atrophic, oligotrophic or normotrophic non-union.	Culture-expanded autologous MSCs combined with biphasic calcium phosphate (BCP) biomaterial granules vs. autograft
A Pre-market, Multi-center, International, Open-label, Single-arm Study to Evaluate the Safety and Performance of a Class III Medical Device (GreenBone Implant) for Surgical Repair of Long Bone Defects	NCT03884790	Unknown	n.a.	Treatment of long bone defects up to 6 cm	GreenBone (ceramic resorbable acellular scaffold)
A Multicenter Study to Evaluate the Enhancement of Bone Regeneration and Healing in the Extremities by the Use of Autologous BonoFill-II	NCT03024008	recruiting	2	Long and short bones extra and intra articular defect/gap or non-union, incapable of self-regeneration	BonoFill-II: tissue-engineered, bone graft consisting autologous adipose tissue-derived mesenchymal stem cells, attached to HA particles.
Safety and Efficacy Study of Traumatic Bone Defects Treatment with Use of 3D Tissue Engineered Equivalent.	NCT03103295	unknown	2	Critical sized long bone defects	3D Tissue Engineered Bone Equivalent: allogeneic or xenogeneic partially demineralized bone matrix and plasma-derived fibrin gel seeded with autologous cultured bone marrow-derived multipotent mesenchymal stromal cells, periosteal progenitor cells, peripheral blood-derived endothelial progenitor cells.
A Phase IIa, Single Center, Prospective, Randomized, Parallel, Two-arms, Single-dose, Open-label With Blinded Assessor Pilot Clinical Trial to Assess ex Vivo Expanded Adult Autologous Mesenchymal Stromal Cells Fixed in Allogeneic Bone Tissue (XCEL-MT-OSTEO-ALPHA) in Non Hypertrophic Pseudoarthrosis of Long Bones	NCT02230514	completed	2a	Non-hypertrophic pseudoarthrosis of long bones	ex vivo expanded adult autologous mesenchymal stromal cells fixed in allogeneic bone tissue (XCEL-MT-OSTEO-ALPHA) vs. autograft
Effect of Bone Marrow-derived Mesenchymal Stem Cell Transplantation in Reconstructing Human Bone Defects	NCT01206179	Completed	1	Non-union or delayed union >4 cm distance to joint	Injection of mesenchymal cells in fracture zone
Potency of Allogenic Bone Marrow, Umbilical Cord, Adipose Mesenchymal Stem Cell for Non Union Fracture and Long Bone Defect, Directly and Cryopreserved	NCT02307435	unknown	1	Non-nion fracture and metaphyseal fibrous defect	allogenic mesenchymal stem cells from umbilical cord/bone marrow/adipose combined with HA-CaSo4

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
