# Peer review of "Scaffold Guided Bone Regeneration for the Treatment of Large Segmental Defects in Long Bones"

_biomedicines, 2023, doi:10.3390/biomedicines11020325_

Round 1

Reviewer 1 Report

This narrative research is under the scope of this journal; the topic is relevant for readers, and this research deals with potentially significant knowledge to the field. And It will be important to different areas of knowledge. 

However, there are some aspects which is possibly improved in the manuscript:

- Correct some typos.

(Keywords)

  • Order the keywords / Mesh Terms alphabetically

Introduction

The topic is relevant for readers and this review deals with potentially significant knowledge to the field and open new way for future studies.

Introduction

  • Is the space provision a limitation for some bone graft material. Regeneration bone defects with scaffolds of the pores or the space provision versus compacts materials, please read this article, please read  (Palma, P. J., Matos, S., Ramos, J., Guerra, F., Figueiredo, M. H., & Krauser, J. (2010). New formulations for space provision and bone regeneration. Biodental Eng. I1, 71-76. WOS:000282776500012; SBN 978-0-415-57394-8) reported the influece of different formulations of bone grafts in providing an adequate scaffold, thus emphasizing the importance of the type of carrier in the three-dimensional distribution of particles and  also space provision in new bone formation. Please, read also this article https://doi.org/10.3390/molecules26051339 for support the bone tissue graft. Also, bone tissue and cementum share many similarities: Bone remodelling represents the most remarkable bone response to mechanical stress and mineral homeostasis. It is the consequence of complex highly orchestrated and tightly regulated cellular processes taking place in a specialized entity - the Bone Remodelling Compartment (BRC). Please read the Brochado Martins, 2020 (Folia Morphologica Journal, 10.5603/FM.a2020.0134), in Remodelling compartment in root cementum (RCRC), Hypothesizing that similar cellular mechanisms underlie bone and cementum remodelling, the present work shows, for the first time, the histological evidence of a specialized remodelling compartment in dental hard tissues (Please add also in discussion).
  • Also described the diferences between  bone fillers and bone substitutes. 
  • This synthetic grafts  will work as  bone fillers or a  bone substitutes?
  • Improve the resolution quality of all figures and graphs (and a presentation). The font/language in the figure/caption is different from the text. Please, standardise the size and the font in the figures and charts with the font of the manuscript. 

Author Response

Reviewer 1

This narrative research is under the scope of this journal; the topic is relevant for readers, and this research deals with potentially significant knowledge to the field. And It will be important to different areas of knowledge.

However, there are some aspects which is possibly improved in the manuscript:

Correct some typos

The manuscript was proofread by all authors and typos and grammar were corrected throughout.

Order the keywords / Mesh Terms alphabetically

Keywords were ordered alphabetically (line 26).

Introduction

The topic is relevant for readers and this review deals with potentially significant knowledge to the field and open new way for future studies.

Introduction

Is the space provision a limitation for some bone graft material. Regeneration bone defects with scaffolds of the pores or the space provision versus compacts materials, please read this article, please read  (Palma, P. J., Matos, S., Ramos, J., Guerra, F., Figueiredo, M. H., & Krauser, J. (2010). New formulations for space provision and bone regeneration. Biodental Eng. I, 1, 71-76. WOS:000282776500012; SBN 978-0-415-57394-8) reported the influece of different formulations of bone grafts in providing an adequate scaffold, thus emphasizing the importance of the type of carrier in the three-dimensional distribution of particles and  also space provision in new bone formation. Please, read also this article https://doi.org/10.3390/molecules26051339 for support the bone tissue graft. Also, bone tissue and cementum share many similarities: Bone remodelling represents the most remarkable bone response to mechanical stress and mineral homeostasis. It is the consequence of complex highly orchestrated and tightly regulated cellular processes taking place in a specialized entity - the Bone Remodelling Compartment (BRC). Please read the Brochado Martins, 2020 (Folia Morphologica Journal, 10.5603/FM.a2020.0134), in Remodelling compartment in root cementum (RCRC), Hypothesizing that similar cellular mechanisms underlie bone and cementum remodelling, the present work shows, for the first time, the histological evidence of a specialized remodelling compartment in dental hard tissues (Please add also in discussion).

We added information about the use of xenogenic bone graft materials. In addition, we added references as suggested by the reviewer (line 243 – 250). Since the current manuscript deals with scaffold-based bone substitutes for treatment of large segmental defects and not dental applications or the use of cementum based void filling, the corresponding commentary of the reviewer was not incorporated. To better clarify the scope of our review and in response to reviewer 2 we changed the title accordingly and included specification of segmental long bone defects, if applicable, throughout the text.

Also described the diferences between bone fillers and bone substitutes. 

This synthetic grafts will work as bone fillers or a bone substitutes?

To address both of these comments, we added a clarification that this review is focused on scaffold-based biomimetic bone substitutes and also provided a definition of this term. In consequence bone void fillers are not the main subject of the manuscript (line 177-180).

Improve the resolution quality of all figures and graphs (and a presentation). The font/language in the figure/caption is different from the text. Please, standardise the size and the font in the figures and charts with the font of the manuscript.

We changed the fonts in all figures to match with the font used in the text. We increased all font sizes when applicable. Resolution of all figures was improved.

Reviewer 2 Report

Schulze et al. They present an interesting manuscript and an important topic. These types of reviews are important for a constantly changing field of research of clinical interest. However, the authors must make a substantial improvement to the manuscript:

-First of all, the authors must include in the title of the manuscript the field and/or area of action of the clinical specialty to which it is addressed.

-The abstract of the manuscript is very simple, and should include the novelty of the manuscript: e.g. updating, integrative vision.....

-It is important that the references are current. All references must be after the year 2019, only a few references can be before.

-There are numerous very long paragraphs with only one reference. This needs to be improved.

-Point 2 is very simple. Authors should include aspects of novelty in this regard, I suggest authors focus on the role of epigenetics, etc... Authors should include the role of MMP/TIMP.

-The authors talk about the role of the extracellular matrix. Please, authors must include the paper of LOX, LOXL-1, FBLN-4.

-Authors must include a more representative figure in section 2.

-Point 3 is very simple. The authors must include aspects of novelty, such as bioprinting, biocompatibility, main preclinical models, substitutes, etc...

-Authors must include a table with the most widespread preclinical models.

-Authors must include a specific section of clinical trials, in all phases (Include a table).

-Authors must include a section on the main biomaterials, with adequate historical and clinical evaluation.

-Figure 4 is meaningless, it is simply informative. Please restructure it.

-Section 5 is not very adequate. The authors must include the novelty.

Author Response

Reviewer 2

Schulze et al. They present an interesting manuscript and an important topic. These types of reviews are important for a constantly changing field of research of clinical interest. However, the authors must make a substantial improvement to the manuscript:

-First of all, the authors must include in the title of the manuscript the field and/or area of action of the clinical specialty to which it is addressed.

We agree with the reviewer that our initial title was too unspecific. We therefore changed the title to specify the scope of the review as following:

Scaffold guided bone regeneration for the treatment of large segmental defects in long bones

-The abstract of the manuscript is very simple, and should include the novelty of the manuscript: e.g. updating, integrative vision.....

We included a specification of the reviews’ scope (line 17 – 18)

-It is important that the references are current. All references must be after the year 2019, only a few references can be before.

We thank the reviewer for this comment and agree that some references were outdated. Therefore, if applicable, references were updated throughout the entire manuscript.

In addition, the last two chapters focusing on preclinical and clinical studies and approaches should be focused on more recent developments. However, we are also discussing and summarizing fields which are still highly challenging (e.g. cell therapy, biomaterials, growth factors) and we are certain that it is important to highlight also studies which have been performed before 2019 to provide an insightful and more complete overview on the field. 

-There are numerous very long paragraphs with only one reference. This needs to be improved.

We addressed this issue throughout the manuscript.

-Point 2 is very simple. Authors should include aspects of novelty in this regard, I suggest authors focus on the role of epigenetics, etc... Authors should include the role of MMP/TIMP.

We added a section on MMPs/TIMPs (line 123 – 147).

-The authors talk about the role of the extracellular matrix. Please, authors must include the paper of LOX, LOXL-1, FBLN-4.

We added a paragraph on LOX proteins and their role in the ECM (line 148 – 172).

-Authors must include a more representative figure in section 2.

Figure 1 was revised and improved.

-Point 3 is very simple. The authors must include aspects of novelty, such as bioprinting, biocompatibility, main preclinical models, substitutes, etc...

We clarified that the manuscript is focused on the use of scaffold-based bone substitutes and provided a definition of the term (line 177 -178).

We added a section on bioprinting (line 465 – 488).

We added additional information on the subject of biocompatibility (line 190 – 193)

Preclinical models are already subject of the manuscript (see 4.1. Large animal models). Section 3 is meant to inform about biomaterials that can be employed while briefly describing their respective advantages and limitations to enable the reader to follow later sections on their preclinical and clinical application.

-Authors must include a table with the most widespread preclinical models.

We thank the reviewer for this valuable suggestion and created a table listing selected studies testing new scaffold-based approaches in full-thickness long bone defects in large animal models between 2018 and 2022. Since the review focuses on translational potential, we feel that it is appropriate to focus such a table on large animal models with great translational potential while results from small animal models are integrated in the chapters focusing on specific subtopics (growth factors etc). Preclinical models are summarized in table 1.

-Authors must include a specific section of clinical trials, in all phases (Include a table).

A table summarizing ongoing clinical trials is now provided as table 2.

-Authors must include a section on the main biomaterials, with adequate historical and clinical evaluation.

Section 3 already lists the most important biomaterials used in the context of scaffold-based treatment of large segmental defects. Historical evaluation of each material category would exceed the scope of the manuscript. Evaluation of materials that have been successfully translated into pre-clinical and clinical application is the subject of section 4 and therefore already part of the manuscript. To better summarize the focus of the materials section, we provided a small preamble to this paragraph (line 227-231).

-Figure 4 is meaningless, it is simply informative. Please restructure it.

In our opinion, vascularization strategies are major preclinical and clinical challenges in large segmental defects of long bones and it is therefore of importance to address this topic with a certain depth in this review. Since several different approaches exist to revascularize bone, we think that a graphical overview on the different approaches is a helpful tool for the reader to easily follow the explanations and structure of the chapter. Moreover, graphical illustrations support especially visual learners and highlight the variety of the different vascularization strategies.

-Section 5 is not very adequate. The authors must include the novelty.

We agree with the reviewer that section 5 was missing novel concepts and the discussion of their relevance for future clinical applications. We therefore improved this section accordingly (line 856 – 868).

Round 2

Reviewer 1 Report

the authors improve the article with review

Reviewer 2 Report

The authors have perfectly answered the reviewers' questions. The manuscript is ready for publication.